# Controllable Attack and Improved Adversarial Training in Multi-Agent Reinforcement Learning

**Xiangyu Liu**
xyliu999@umd.edu
University of Maryland, College Park

**Souradip Chakraborty**
schakra3@umd.edu
University of Maryland, College Park

**Furong Huang**
furongh@umd.edu
University of Maryland, College Park

## Abstract

Deep reinforcement learning policies have been shown vulnerable to adversarial attacks due to the inherent frangibility of neural networks. Current attack methods mainly focus on the adversarial state or action perturbations, where such direct manipulations to a reinforcement learning system may not always be feasible or realizable in the real-world. In this paper, we consider the more practical adversarial attacks realized through actions by an adversarial agent in the same environment. It has been shown, in prior work, that a victim agent is vulnerable to behaviors of an adversarial agent who targets to attack the victim, at the cost of introducing perceivable abnormal behaviors for the adversarial agent itself. To address this, in the first part of this paper, we propose to constrain the state distribution shift caused by the adversarial policy and offer a more controllable attack scheme by building connections among policy space variations, state distribution shift, and the value function difference. To provide provable defense, we revisit the cycling behavior of common adversarial training methods in Markov game, which has been a well-known issue in general differential games including Generative Adversarial Networks (GANs) and adversarial training in supervised learning. We propose to fix the non-converging behavior through a simple timescale separation mechanism. In sharp contrast to general differential games, where timescale separation may only converge to stationary points, two-timescale training methods in Markov games can converge to the Nash Equilibrium (NE). Using the Robosumo competition experiments, we demonstrate the controllable attack is much more efficient in the sense that it can introduce much less state distribution shift while achieving the same winning rate with unconstrained attack. Furthermore, in both Kuhn Poker and Robosumo competition, we verify that the rule of timescale separation leads to stable learning dynamics and less exploitable victim policies.

## 1 Introduction

Despite the huge success of deep reinforcement learning (RL) algorithms across various domains [22, 16, 20], it has been shown that deep reinforcement learning policies are highly vulnerable to adversarial attacks. The most popular attack methods focus on fooling the RL agent by adversarially perturbing the states or observations. The success of such attacks can be attributed to lack of adversarial robustness in the deep neural network which is used to parametrize the value functions/Q-functions or the policies of the deep RL algorithms. One of the earliest work by [9] showed a detailed study of various adversarial attacks on the neural network policies and developed threat

2022 Trustworthy and Socially Responsible Machine Learning (TSRML 2022) co-located with NeurIPS 2022.

models capable of significantly degrading the test-time performance of trained policies with minimal perturbation. [13] emphasized on enhancing the efficiency of the adversarial attack by minimizing the agent's reward by only attacking the agent at a small subset of time steps in an episode and in the most significant times. There has been subsequent research in developing more efficient pixel-based attacks [19, 18, 17]. [26] built the theoretical framework SA-MDP for adversarial state perturbation and proposed the corresponding regularizer for more robust reinforcement learning policies. Subsequent work [24] improved [26] with the framework of PA-MDP for better efficiency.

However, perturbing states or observations of an agent might not always be feasible/practical in real-life scenarios such as self-driving cars and robotic manipulation, where manipulating the observations may require attacking the communication systems which is hard to achieve. Consequently, in this paper, we consider the attacks realizable through actions of an adversarial agent in the same environment as the victim. Along this direction, [6] follows a principled approach towards attacking well-trained RL agents by training adversarial policies of an opponent agent to minimize the expected return of the victim. The adversarially trained policy efficiently defeats the state-of-the-art agents trained via self-play [2] despite the adversarial policy being trained for less than 3% of the training time steps. In zero-sum games, the goal to maximize the expected return of the attacker coincides with minimizing the expected return of the victim. For games, which are not strictly zero-sum, [7] proposed to maximize the expected return of the attacker and minimize the expected return of the victim simultaneously and improved the attack efficiency. If the attacker is allowed to access the observations and actions of the victim, [25] showed that fitting the environment transition model and the victim policy model can help improve the attack success rate.

However, while seeking the policy to degrade the performance of the victim, the attacker is usually optimized in an unconstrained fashion thereby resulting in adversaries demonstrating abnormal behaviours [6], which could be easily detected by human eyes and thus not consistent with the spirit of imperceivable adversarial examples in supervised learning. To address this, we propose to constrain the state distribution shift caused by the adversarial policy and offer a more controllable attack scheme by building connections among policy space variations, state distribution shift, and the value function difference. On the other hand, to promote robustness of the victim policy, we investigate the adversarial training methods and verify the diverging behavior in Markov games, which has been a known issue in general differential games including Generative Adversarial Networks (GANs) [3, 8, 5] and adversarial training in supervised learning. To fix this, we introduce a simple timescale separation mechanism, which is guaranteed to converge to Nash Equilibrium (NE) with tabular parameterization. To summarize, this work makes contributions to both attack and defense in the multi-agent reinforcement learning. For attack, we propose a controllable attack framework to mitigate the large state distribution shift introduced by the unconstrained attack. For defense, we revisit the non-converging behaviors of the adversarial training even in Markov games and introduce the simple rule of timescale separation, which enjoys both strong theoretical guarantee (converging to NE standing in contrast to local convergence in differential games) and leads to stable training dynamics and much less exploitable victim policy.

## 2   Preliminaries

The extension of Markov decision processes (MDPs) with more than one agents is commonly modelled as Markov games [14]. A Markov game with $N$ agents is defined by a tuple $< N, \mathcal{S}, \{\mathcal{A}_i\}_{i=1}^N, P, \{r_i\}_{i=1}^N, \rho, \gamma >$, where $\mathcal{S}$ denotes the state space and $\mathcal{A}_i$ is the action space for agent $i$. The function $P$ controls the state transitions by the current state and one action from each agent: $P : \mathcal{S} \times \mathcal{A}_1 \times \cdots \times \mathcal{A}_N \to \Delta(\mathcal{S})$, where $\Delta(\mathcal{S})$ denotes the set of probability distributions over the state space $\mathcal{S}$. Given the current state $s_t$ and the joint action $(a_1, \ldots, a_N)$, the transition probability to $s_{t+1}$ is given by $P(s_{t+1}|s_t, a_1, \ldots, a_N)$. The initial state is sampled from the initial state distribution $\rho \in \Delta(S)$. Each agent $i$ also has an associated reward function $r_i : \mathcal{S} \times \mathcal{A}_i \times \cdots \times \mathcal{A}_N \to [0, 1]$. Each agent's goal is to maximize the $\gamma$-discounted expected return $R_i = \mathbb{E}[\sum_{t=0}^\infty \gamma^t r_i(s_t, a_i^t, a_{-i}^t)]$, where $-i$ is a compact representation of all complementary agents of $i$. In the following discussions, we mainly focus on the two-player zero-sum games, where we have two agents labeled as $\alpha$ and $\mu$ and the reward satisfies $r_\alpha + r_\nu = 0$.

In Markov games, each agent is equipped with a policy $\pi_i : \mathcal{S} \to \Delta(\mathcal{A}_i)$. The corresponding policy class is denoted as $\Pi_i$ and the joint policy is defined as $\boldsymbol{\pi}(\mathbf{a}|s) = \Pi_{i=1}^N \pi_i(a_i|s)$. The value function for the two-player zero-sum game given joint policy $(\pi_\alpha, \pi_\nu)$ is defined by $V_s(\pi_\alpha, \pi_\nu) = \mathbb{E}_{\pi_\alpha, \pi_\nu} \left[ \sum_{t=0}^\infty \gamma^t r_\alpha(s_t, \mathbf{a}_t) \mid s_0 = s \right]$, where player $\alpha$ attempts to maximize the value function and

player $\nu$ minimizes this value. We abuse the notation $V_\rho(\pi_\alpha, \pi_\nu) := \mathbb{E}_{s \sim \rho}[V_\rho(\pi_\alpha, \pi_\nu)]$. We further define state visitation, which reflects how often the policy visits different states in the state space.

**Definition 1.** *(Stationary State Visitation) Let $d_{\boldsymbol\pi} \in \Delta(\mathcal{S})$ denote the normalized distribution of state visitation by following the joint policy $\boldsymbol\pi$ in the environment:*

$$d_{\boldsymbol\pi}(s) = (1 - \gamma) \sum_{t=0}^{\infty} \gamma^t P(s_t = s | \boldsymbol\pi) . \tag{1}$$

## 3 Controllable Adversarial Attack

The authors of [6] consider the more realistic attack model in which attacks are realized through actions of an adversarial agent in the same environment as the victim. Although principled, the attacking formula may be too strong as the attacks may no longer be stealthy or imperceptible. People may argue that once the attacks are non-stealthy, they can be easily recognized and thus no longer be harmful. Here, we make one essential hypothesis that humans usually detect the presence of adversarial behaviors by tracking the states the multi-agent systems. Therefore, for the adversarial agent to hide the adversarial behaviors from being detected, we propose to control the perturbation of the system states. Formally, given a multi-agent system with policies $(\pi_\alpha, \pi_\nu)$, we offer our attack objective as the following optimization problem.

$$\min_{\pi'_\nu} V_\rho(\pi_\alpha, \pi'_\nu) \tag{2}$$

$$\text{s.t.} \quad ||d_{\pi_\alpha, \pi_\nu} - d_{\pi_\alpha, \pi'_\nu}||_1 \leq \epsilon, \tag{3}$$

The objective (2) is a common attack objective, where the adversarial player aims to minimize the value of the victim. Apart from the objective, we propose further to constrain the behavior of the adversarial policy $\pi'_\nu$ so that the induce state visitation of $d_{\pi_\alpha, \pi'_\nu}$ stays close to the original $d_{\pi_\alpha, \pi_\nu}$. However, it is generally difficult to estimate the state visitation for given policies, let alone enforce the constraints in (3). Therefore, in the remaining section, we will try to find an upper bound for the discrepancy between state visitation. The key idea is to show that the distance between state visitation can be upper bounded by the distance between policies. To do so, we firstly define the distance between two policies as follows

**Definition 2** (Attack Budget). *We define the maximum total variation distance $D_{\mathrm{TV}}^{\max}$ between two policies as: $D_{\mathrm{TV}}^{\max}(\pi || \pi') = \max_s D_{\mathrm{TV}}(\pi(\cdot|s) || \pi'(\cdot|s))$, where $D_{\mathrm{TV}}(p || q) = \frac{1}{2} \sum_i |p_i - q_i|$.*

This definition gives a more straight-forward way to control the variation of the policy at every state $s$. However, it is unclear how this definition relates to the total variation between state visitations. We will answer this in the following theorem.

**Theorem 1.** *For two victim-attacker policy pairs $(\pi_\alpha, \pi_\nu)$ and $(\pi_\alpha, \pi'_\nu)$, where $D_{\mathrm{TV}}^{\max}(\pi_\nu || \pi'_\nu) \leq \epsilon$, the difference between the state visitation of $(\pi_\alpha, \pi_\nu)$ and $(\pi_\alpha, \pi'_\nu)$ can be bounded as: $||d_{\pi_\alpha, \pi_\nu} - d_{\pi_\alpha, \pi'_\nu}||_1 \leq \frac{2\gamma\epsilon}{1-\gamma}$.*

This theorem shows a relationship between $D_{\mathrm{TV}}^{\max}(\pi_\nu || \pi'_\nu)$ and $||d_{\pi_\alpha, \pi_\nu} - d_{\pi_\alpha, \pi'_\nu}||_1$: to control the variation of the state visitation, it suffices to control the policy variation at each state $s$. Meanwhile, with the state distribution variation controlled, we derive the following robustness certificate:

**Corollary 1** (Robust Certificate). *Under the same condition of Theorem 1, the following value difference holds $|V_\rho(\pi_\alpha, \pi_\nu) - V_\rho(\pi_\alpha, \pi'_\nu)| \leq \frac{2\epsilon}{(1-\gamma)^2}$.*

This corollary suggests that the value function in Markov games naturally satisfies a Lipschitz condition, where the attacker must have enough budget to achieve large performance degradation, standing in sharp contrast with the adversarial attack in supervised learning, where small perturbations suffices to make huge changes to the final outputs. Formally, this tells us the performance drop is at most $\frac{2\epsilon}{(1-\gamma)^2}$ when the attacker has the attack budget $\epsilon$.

**A parameterization that converts the constrained problem to unconstrained.** As shown in Theorem 1, policy variation upper-bounds the state distribution variation. Consequently, rather than directly constrain state visitation distribution, we constrain policy variation $D_{\mathrm{TV}}^{\max}(\pi_\nu || \pi'_\nu)$, which can be realized more efficiently and sufficiently. We use the idea from conservative policy iteration [10] and convert the constrained problem to the unconstrained by using the following parameterization.

**Definition 3** ($\epsilon$ coupling). *The $\epsilon$ coupled policy is constructed for each state $s \in \mathcal{S}$*

$$\pi'_\nu(\cdot \mid s) = (1 - \epsilon)\pi_\nu(\cdot \mid s) + \epsilon\pi_\nu(\theta)(\cdot \mid s), \tag{4}$$

*where $\pi_\nu$ is the original attacker, which is fixed during attack and $\pi_\nu(\theta)$ parameterized by $\theta$ is what the adversary tries to optimize.*

With $\epsilon$ coupling, it is easy to verify the following proposition.

**Proposition 1.** *With $\pi'_\nu$ defined as (4), it holds for all $\theta$ that $D_{\mathrm{TV}}^{\max}(\pi_\nu||\pi'_\nu) \leq \epsilon$.*

Therefore, the objective (2) can be replaced by $\min_\theta V_\rho(\pi_\alpha, (1 - \epsilon')\pi_\nu + \epsilon'\pi_\nu(\theta))$, where $\epsilon' = \frac{1-\gamma}{2\gamma}\epsilon$ according to theorem 1 and proposition 1.

## 4 Improved Adversarial Training with Timescale Separation

Considering the vulnerability of victim policy, effective defense strategies are in urgent need. There have been prior work on utilizing the attack for re-training to ensure the robustness of the victim policy [6, 7, 25]. However, it has been shown that although re-training against a specific adversarial policy provides robustness against the specific attacker, the performance against benign policies is actually degraded. Therefore, it is important to ensure the performance of victim against all possible attackers. and adversarial attacker. To begin with, in this work, we will start from providing defense for the basic unconstrained attack, i.e. $\epsilon = 1$, for which we will see common adversarial training methods suffer from the non-converging problem. We leave the defense to attack of different intensities as future works. Since we only consider the robustness of the victim, we define the exploitability of $\pi_\alpha$ using the following one-side exploitability.

**Definition 4** ((One-side) exploitability). *For a victim policy $\pi_\alpha$, we meaure the robustness of $\pi_\alpha$ by:* $\mathrm{Expl}(\pi_\alpha) = -\min_{\pi_\nu} V_\rho(\pi_\alpha, \pi_\nu)$.

To ensure the worst-case performance against the strongest adversarial policy, the victim should choose the policy according to $\min_{\pi_\alpha} \mathrm{Expl}(\pi_\alpha)$, which is equivalent to $\max_{\pi_\alpha}\min_{\pi_\nu} V_\rho(\pi_\alpha, \pi_\nu)$, coinciding with finding the Nash Equilibrium in zero-sum Markov games [21]. To motivate the necessity of timescale separation in adversarial training for robust RL policy, we revisit some known issues of naive methods including Gradient Descent Ascent (GDA) and iterative best response (IBR) with both simultaneous and alternate update using a simple normal-form game Rock-Paper-Scissor, which is a Markov game with one step. For detailed comparison, we investigate four single timescale algorithm and one algorithm with timescale separation in Appendix A. Formally, we propose to improve the adversarial training via timescale separation with Min oracle (shown in algorithm 1), where the attacker takes a min step against the victim in line 3 while the victim takes one gradient update in line 4. The min oracle used in algorithm 1 can be implemented with standard reinforcement

---

**Algorithm 1** Adversarial Training with Min-oracle

1: **Input:** random policy $\pi_\alpha^0$, learning rate sequence $\{\eta^t\}$
2: **for** $t = 1$ to $T$ **do**
3:     $\pi_\nu^t \leftarrow \arg\min_{\pi_\nu} V_\rho(\pi_\alpha^t, \pi_\nu)$.
4:     $\pi_\alpha^{t+1} \leftarrow \mathcal{P}_{\Pi_\alpha}(\pi_\alpha^t + \eta^t\nabla_{\pi_\alpha}V_\rho(\pi_\alpha^t, \pi_\nu^t))$.
5: **end for**
6: **Output:** sample $\pi_\alpha^t$ with probability proportional to $\eta^t$.

---

learning algorithm. When the game has special structures like extensive-form games or one player has substantially smaller state and action space, such min oracle may be implemented efficiently. In general, to make one gradient update for player $\alpha$, agent $\nu$ needs to solve the corresponding best response oracle, which is computationally inefficient in practice. To further fix this issue, we utilize the idea of using a much larger step size for player $\nu$ so that when player $\alpha$ performs the gradient update, player $\nu$ is already an approximate minimum of $V_\rho(\pi_\alpha^t, \cdot)$. Formally, in addition to Algorithm 1, we present an alternative efficient Algorithm 2, where the min oracle is replaced by a gradient update with a larger step and both players perform gradient update independently. Theoretically, the algorithm is guaranteed to converge to (one-side) Nash Equilibrium by directly adopting the main conclusion of Theorem 1 from [4]. Formal statements are given in Appendix B.

# 5 Experimental Setup and Results

Our experiments are based on two common zero-sum games Kuhn Poker [11] and RoboSumo competition [1]. Detailed introductions to the environment are given in Appendix D. In this section, we seek answers to the following two essential questions.

▷ (1) Can the constrained attack formulation yield less state distribution variations (being more imperceptible) compared with unconstrained attack with the same winning rate?

▷ (2) Will adversarial training with time scale separation achieve more stable learning dynamics and better/less robustness/exploitability compared with only single time scale?

**Controllable Adversarial Attack.** To answer our first question, we conduct experiments on the Robosumo Spider vs Spider environment. To verify that our constrained attack approach indeed achieves smaller state distribution variations, we fix an unconstrained adversarial policy and a constrained policy with the same winning rate for fair comparisons. Specifically, we investigate the distribution shift in the victim's observation part of the state features. To select important state features, we use the variance-based feature importance method to filter out the unimportant features of the states with small variance. The feature variance is determined by observing the change in the features over several episodes. The sorted feature importance is given in Appendix D.2 for the Robosumo Spider-vs-Spider game. Figure 1 shows that the constrained adversarial policy induces much smaller state distribution shift while compared to the unconstrained adversarial policy when testing under the same winning rate.

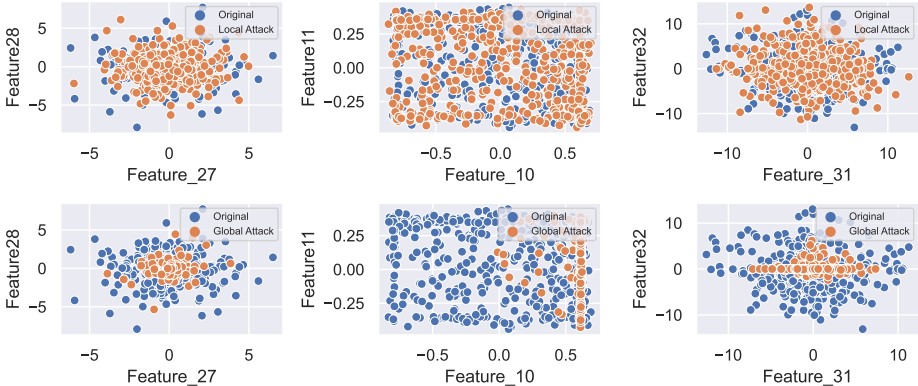

Figure 1: Visualization and comparison of our proposed constrained attack with $\epsilon = 0.2$ (first row) vs an unconstrained attack (second row), under the condition that both achieve the same attacking success rate. State feature distributions of victim's observation for most critical features are illustrated. It is clearly that our constrained adversarial policy induces much smaller state distribution shift.

**Improved Adversarial Training.** To answer our second question, we compare the learning dynamics and robustness of policies trained by two time scales and single time scales in both Kuhn Poker and Robosumo Ant-vs-Ant environments.

*Kuhn Poker.* We implement both our algorithm 1 and 2 with open-source project OpenSpiel [12], where the min oracle is achieved by search in the game tree. For gradient update, instead of using the vanilla monte-carlo policy gradient, we use both Regret Policy Gradient (RPG) [23] and Advantage Actor Critic (A2C) [15] methods. In figure 2, we show exploitability of the victim policy $\pi_\alpha$ averaged over 5 independent seeds, where adversarial training with the min oracle enjoys the fastest convergence, lowest exploitability and variance. Meanwhile, the victim policy trained with large enough timescale separation parameter $\eta_\nu^t / \eta_\alpha^t$ approximate the algorithm with a min oracle. On the contrary, adversarial policy trained with only single timescale suffers from high exploitability and variance even when the step size is small enough.

*Robosumo Competition.* To verify the necessity of timescale separation, apart from the previous relatively simple Kuhn Poker experiments, we also evaluate our proposed methods on Robosumo Ant vs Ant, a high-dimensional, continuous control task, which is much more challenging in terms of both training and evaluation. For training, although such a min oracle with game tree search in the Kuhn Poker does not exist in this continuous control task anymore, previous experiments highlight that a large enough timescale separation ratio can very well approximate the algorithm 1 with a min oracle.

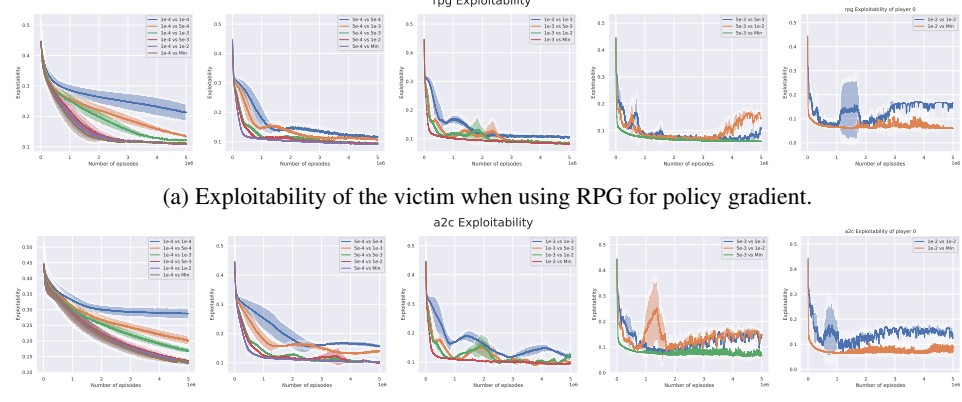

(a) Exploitability of the victim when using RPG for policy gradient.

(b) Exploitability of the victim when using A2C for policy gradient.

Figure 2: Exploitability of victim policy in Kuhn Poker trained by two timescale and single timescale (min indicates the policy trained with a min oracle). The error bar calculated by 5 independent seeds.

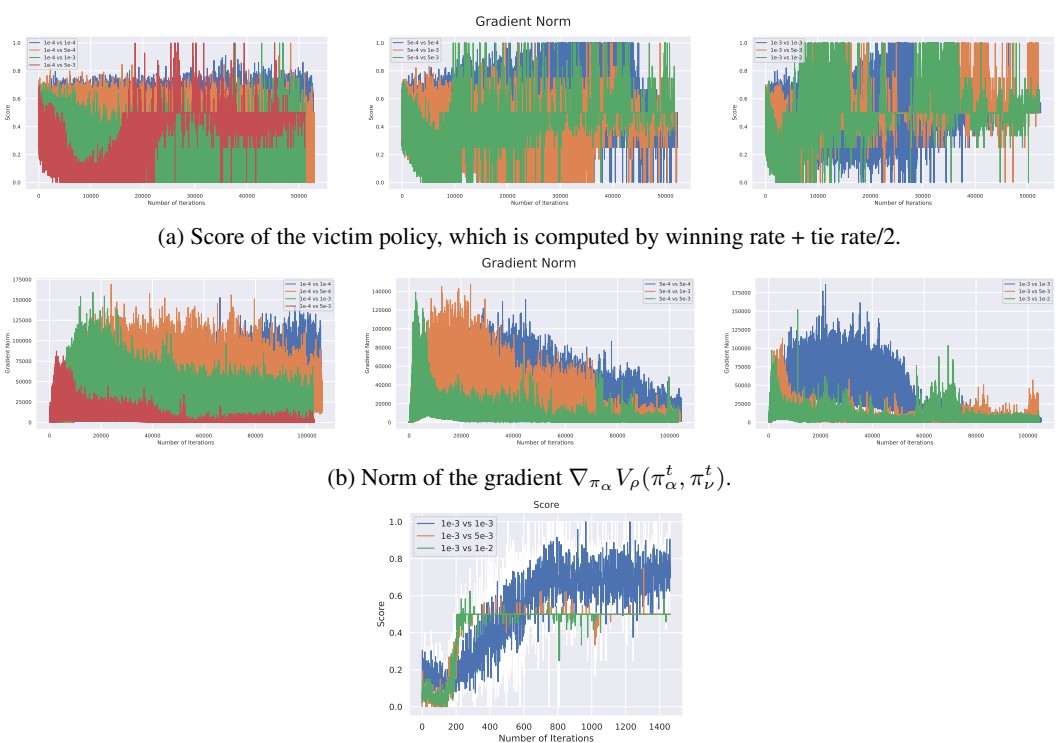

(a) Score of the victim policy, which is computed by winning rate + tie rate/2.

(b) Norm of the gradient $\nabla_{\pi_\alpha} V_\rho(\pi_\alpha^t, \pi_\nu^t)$.

(c) Score of the attacker against robustified policy (final output of adversarial training). Lower is better.

Figure 3: For experiments on the Robosumo Ant vs Ant, we report both the score and the gradient norm of the victim policy. (a). For score, since this game is symmetric, the equilibrium value will be exactly $0.5$, where score of victim policy trained by two timescale converges to this equilibrium value, while the policy trained only with single timescale suffers from much more oscillations. (b). Gradient norm trained by two time scale is also much smaller. (c). When attacking the robustified victim policy, i.e. computing $\min_{\pi_\nu} V_\rho(\pi_\alpha^\star, \pi_\nu)$ with standard RL algorithm, victim trained by two timescale achieves the lowest/best exploitability/robustness.

For evaluation, again due to the lack of an efficient min oracle, we can not track the exploitability of the victim at every round of training. Instead, we use standard RL algorithm to compute the best response of the final robustified victim policy and keep track of the score of the attacker in figure 3c, where lower winning rate for the attacker indicates better robustness of the victim. Apart from the exploitability test, we also keep track of the score, $V_\rho(\pi_\alpha^t, \pi_\nu^t)$ during training and the norm of gradient $\nabla_{\pi_\alpha} V_\rho(\pi_\alpha^t, \pi_\nu^t)$. From figure 3a and 3b, we verify that single timescale training leads to unstable behaviors, large variance and gradient norm, while the two timescale training leads to quickly converged value and much smaller gradient norm. On the other hand, victim policy trained by two timescale shows much better robustness when the attacker computes the best response against it.

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

# Controllable Attack and Improved Adversarial Training in Multi-Agent Reinforcement Learning"

## A   An Illustrative Example

**Example 1.** *The zero-sum game Rock-Paper-Scissor includes two players with the same action space* $\mathcal{A} = \{Rock, Paper, Scissor\}$. *The payoff matrix* $\mathbf{P}$ *is given as* $\mathbf{P} = \begin{bmatrix} 0 & 1 & -1 \\ -1 & 0 & 1 \\ 1 & -1 & 0 \end{bmatrix}$ *for the row player. The row player has the mixed strategy* $x \in \mathcal{X} = \Delta(\mathcal{A})$, *where* $x_i$ *represents the probability of choosing* $i^{th}$ *action. The column player holds a similar mixed strategy* $y \in \mathcal{Y}$. *The corresponding payoff is given by* $V(x, y) = x^\top \mathbf{P} y$. *Our objective is given by* $\max_{x \in \mathcal{X}} \min_{y \in \mathcal{Y}} x^\top \mathbf{P} y$.

We evaluate the following 5 methods.

- Simultaneous Gradient Descent Ascent (SGDA): $y_{t+1} = \mathcal{P}_\mathcal{Y}(y_t - \eta \mathbf{P}^\top x_t)$, $x_{t+1} = \mathcal{P}_\mathcal{X}(x_t + \eta \mathbf{P} y_t)$.
- Alternate Gradient Descent Ascent (AGDA): $y_{t+1} = \mathcal{P}_\mathcal{Y}(y_t - \eta \mathbf{P}^\top x_t)$, $x_{t+1} = \mathcal{P}_\mathcal{X}(x_t + \eta \mathbf{P} y_{t+1})$
- Simultaneous Iterative Best Response (SIBR): $y_{t+1} = \arg\min_{y \in \mathcal{Y}} x_t^\top \mathbf{P} y$, $x_{t+1} = \arg\min_{x \in \mathcal{X}} x^\top \mathbf{P} y_t$
- Alternate Iterative Best Response (AIBR): $y_{t+1} = \arg\min_{y \in \mathcal{Y}} x_t^\top \mathbf{P} y$, $x_{t+1} = \arg\min_{x \in \mathcal{X}} x^\top \mathbf{P} y_{t+1}$.
- Gradient Ascent with Min oracle (GAMin): $y_{t+1} = \arg\min_{y \in \mathcal{Y}} x_t^\top \mathbf{P} y$, $x_{t+1} = \mathcal{P}_\mathcal{X}(x_t + \eta \mathbf{P} y_{t+1})$.

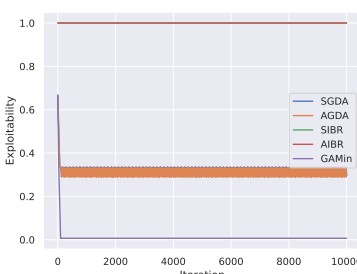

Figure 4: Exploitability test on Rock-Paper-Scissor. Note that blue curve is overlapped with orange and green is overlapped with red.

We show the exploitability of the $x$ player during the learning process in figure 4. It is clear that the first four single timescale training methods (SGDA, AGDA, SIBR, AIBR) fail to achieve low exploitability, while only GAMin achieves the near optimal exploitability 0.

## B   Adversarial Training with Two Timescale

**Theorem 2.** *Fix* $\delta \geq 0$. *Support both* $\pi_\alpha$ *and* $\pi_\nu$ *follows the direct parameterization as* $\pi_\alpha(a_\alpha \mid s) = (1 - \delta_\alpha) x_{s,a_\alpha} + \frac{\delta_\alpha}{|\mathcal{A}_\alpha|}$ *and* $\pi_\nu(a_\nu \mid s) = (1 - \delta_\nu) y_{s,a_\nu} + \frac{\delta_\nu}{|\mathcal{A}_\nu|}$ *and use the single episode REINFORCE gradient estimator. If the step size satisfies* $\eta_\alpha \asymp \delta^{10.5}$, $\eta_\nu \asymp \delta^6$, *and the exploration parameters satisfy* $\delta_\alpha \asymp \delta$, $\delta_\nu \asymp \delta^2$, *it is guaranteed that*

$$\max_{\pi_\alpha} \min_{\pi_\nu} V_\rho(\pi_\alpha, \pi_\nu) - \mathbb{E}\left[\frac{1}{T} \sum_{i=1}^{T} \min_{\pi_\nu} V_\rho(\pi_\alpha^t, \pi_\nu)\right] \leq \delta, \tag{5}$$

*after* $T \leq \mathrm{poly}(\frac{1}{\delta}, |\mathcal{S}|, |\mathcal{A}_\alpha|, |\mathcal{A}_\nu|)$ *iterations.*

## C   Full Proof

### C.1   Proof of Theorem 1

*Proof.* Let us firstly review the following facts for any joint policy $\boldsymbol{\pi} = (\pi_\alpha, \pi_\nu)$, $\boldsymbol{\pi}' = (\pi_\alpha, \pi_\nu')$ with $D_{\mathrm{TV}}^{\max}(\pi_\nu || \pi_\nu') \leq \epsilon$ and the transition matrix $P_{\boldsymbol{\pi}}$, where $P_{\boldsymbol{\pi}}(s', s) = \sum_{\boldsymbol{a}} \boldsymbol{\pi}(\boldsymbol{a}|s) P(s'|s, \boldsymbol{a})$ and we use $P_{\boldsymbol{\pi}}(i, j)$ denote $P_{\boldsymbol{\pi}}(s_i, s_j)$.

---
**Algorithm 2** Adversarial Training with Two Time Scale

---
1: **Input:** random policy $\pi_\alpha^0$, $\pi_\nu^0$, learning rate sequence $\{\eta_\alpha^t\}$, $\{\eta_\nu^t\}$, such that $\eta_\alpha^t \ll \eta_\nu^t$.
2: **for** $t = 1$ **to** $T$ **do**
3:     $\pi_\alpha^{t+1} \leftarrow \mathcal{P}_{\Pi_\alpha}(\pi_\alpha^t + \eta_\alpha^t \nabla_{\pi_\alpha} V_\rho(\pi_\alpha^t, \pi_\nu^t))$.
4:     $\pi_\nu^{t+1} \leftarrow \mathcal{P}_{\Pi_\nu}(\pi_\nu^t - \eta_\nu^t \nabla_{\pi_\nu} V_\rho(\pi_\alpha^t, \pi_\nu^t))$.
5: **end for**
6: **Output:** sample $\pi_\alpha^t$ with probability proportional to $\eta_\alpha^t$.

---

- $d_{\boldsymbol{\pi}} = (1-\gamma)(I - \gamma P_{\boldsymbol{\pi}})^{-1}\rho$.

- $||P_{\boldsymbol{\pi}}||_1 = 1$ and $||(I - \gamma P_{\boldsymbol{\pi}})^{-1}||_1 \leq \frac{1}{1-\gamma}$.

- $||P_{\boldsymbol{\pi}} - P_{\boldsymbol{\pi}'}||_1 \leq 2\epsilon$

According to the definition in Equation 1, one can verify that $d_{\boldsymbol{\pi}}$ satisfies that:
$$d_{\boldsymbol{\pi}} = (1-\gamma)\rho + \gamma P_{\boldsymbol{\pi}} d_{\boldsymbol{\pi}}, \tag{6}$$
which gives the solution $d_{\boldsymbol{\pi}} = (1-\gamma)(I - \gamma P_{\boldsymbol{\pi}})^{-1}\rho$.

For $P_{\boldsymbol{\pi}}$:
$$||P_{\boldsymbol{\pi}}||_1 = \max_j \sum_i |P_{\boldsymbol{\pi}}(i,j)| \tag{7}$$
$$= \max_j \sum_i \sum_{\boldsymbol{a}} \boldsymbol{\pi}(\boldsymbol{a}|s_j)P(s_i|s_j, \boldsymbol{a}) \tag{8}$$
$$= \max_j \sum_{\boldsymbol{a}} \boldsymbol{\pi}(\boldsymbol{a}|s_j) \sum_i P(s_i|s_j, \boldsymbol{a}) \tag{9}$$
$$= 1 \tag{10}$$

For $||(I - \gamma P_{\boldsymbol{\pi}})^{-1}||_1$:
$$||(I - \gamma P_{\boldsymbol{\pi}})^{-1}||_1 = ||\sum_{k=0}^{\infty}(\gamma P_{\boldsymbol{\pi}})^k||_1 \leq \sum_{k=1}^{\infty}||(\gamma P_{\boldsymbol{\pi}})^k||_1 \leq \sum_{k=1}^{\infty}\gamma^k||P_{\boldsymbol{\pi}}||_1^k = \frac{1}{1-\gamma}. \tag{11}$$

For $||P_{\boldsymbol{\pi}} - P_{\boldsymbol{\pi}'}||_1$:

To begin with, since $\pi_\alpha$ remains unchanged in our proof, let us abuse the notation a little bit and define the marginalized transition $P(s'|s, a_\nu) = \sum_{a_\alpha}\pi_\alpha(a_\alpha|s)P(s'|s, a_\alpha, a_\nu)$. We have
$$||P_{\boldsymbol{\pi}} - P_{\boldsymbol{\pi}'}||_1 = \max_j \sum_i |P_{\boldsymbol{\pi}}(i,j) - P_{\boldsymbol{\pi}'}(i,j)| \tag{12}$$
$$= \max_j \sum_i |P_{\boldsymbol{\pi}}(i,j) - P_{\boldsymbol{\pi}'}(i,j)| \tag{13}$$
$$= \max_j \sum_i |\sum_{a_\nu}(\pi_\nu(a_\nu|s_j) - \pi_\nu'(a_\nu|s_j)\sum_{a_\alpha}\pi_\alpha(a_\alpha|s_j)P(s_i|s_j, a_\alpha, a_\nu)| \tag{14}$$
$$= \max_j \sum_i |\sum_{a_\nu}(\pi_\nu(a_\nu|s_j) - \pi_\nu'(a_\nu|s_j))P(s_i|s_j, a_\nu)|. \tag{15}$$

Now fix any index $j$, define $\boldsymbol{m}_i^\top = (P(s_i|s_j, a_\nu^k))_{k=1}^{|\mathcal{A}_\nu|}$, $\boldsymbol{M}^\top = (\boldsymbol{m}_1, \cdots, \boldsymbol{m}_{|\mathcal{S}|})$, and $\boldsymbol{n}^\top = (\pi_\nu(a_\nu^k|s_j) - \pi_\nu'(a_\nu^k|s_j))_{k=1}^{|\mathcal{A}_\nu|}$. Then the following holds
$$\sum_i |\sum_{a_\nu}(\pi_\nu(a_\nu|s_j) - \pi_\nu'(a_\nu|s_j))P(s_i|s_j, a_\nu)| = \sum_i |\boldsymbol{m}_i^\top \boldsymbol{n}| = ||\boldsymbol{Mn}||_1 \leq ||\boldsymbol{M}||_1||\boldsymbol{n}||_1 = 2\epsilon||\boldsymbol{M}||_1. \tag{16}$$

According to the definition of $\boldsymbol{M}$, it is easy to check
$$||\boldsymbol{M}||_1 = \max_k \sum_i |P(s_i|s_j, a_\nu^k)| = 1. \tag{17}$$

Therefore, we conclude that for any fixed index $j$, we have

$$\sum_i |\sum_{a_\nu} (\pi_\nu(a_\nu|s_j) - \pi'_\nu(a_\nu|s_j))P(s_i|s_j, a_\nu)| \leq 2\epsilon, \tag{18}$$

which proves $||P_{\boldsymbol{\pi}} - P_{\boldsymbol{\pi}'}||_1 \leq \epsilon$.

Now we are ready to prove our theorem

$$||d_{\boldsymbol{\pi}} - d_{\boldsymbol{\pi}'}||_1 = ||(1-\gamma)(I - \gamma P_{\boldsymbol{\pi}})^{-1}\rho - (1-\gamma)(I - \gamma P_{\boldsymbol{\pi}'})^{-1}\rho||_1 \tag{19}$$

$$\leq (1-\gamma)||(I - \gamma P_{\boldsymbol{\pi}})^{-1} - (I - \gamma P_{\boldsymbol{\pi}'})^{-1}||_1||\rho||_1 \tag{20}$$

$$\leq (1-\gamma)||I - \gamma P_{\boldsymbol{\pi}'}||_1||\gamma(P_{\boldsymbol{\pi}} - P_{\boldsymbol{\pi}'})||_1||I - \gamma P_{\boldsymbol{\pi}}||_1 \tag{21}$$

$$\leq \frac{2\epsilon\gamma}{1-\gamma}. \tag{22}$$

$\square$

## C.2 Proof of Corollary 1

It is easy to verify that the following holds

$$V_\rho(\boldsymbol{\pi}) = \sum_s d_{\boldsymbol{\pi}}(s) \sum_a \boldsymbol{\pi}(\boldsymbol{a}|s)r(s, \boldsymbol{a}) \tag{23}$$

Let us define the marginalized reward $r_{\boldsymbol{\pi}}(s) = \sum_a \boldsymbol{\pi}(\boldsymbol{a}|s)r(s, \boldsymbol{a})$, and further define the vector notation $r_{\boldsymbol{\pi}}^\top = (r_{\boldsymbol{\pi}}(s^k))_{k=1}^{|\mathcal{S}|}$. Now for the difference of the value function

$$|V_\rho(\boldsymbol{\pi}) - V_\rho(\boldsymbol{\pi}')| = \frac{1}{1-\gamma}|\langle d_{\boldsymbol{\pi}}, r_{\boldsymbol{\pi}}\rangle - \langle d_{\boldsymbol{\pi}'}, r_{\boldsymbol{\pi}'}\rangle| \tag{24}$$

$$= \frac{1}{1-\gamma}|\langle d_{\boldsymbol{\pi}}, r_{\boldsymbol{\pi}}\rangle - \langle d_{\boldsymbol{\pi}}, r_{\boldsymbol{\pi}'}\rangle + \langle d_{\boldsymbol{\pi}}, r_{\boldsymbol{\pi}'}\rangle - \langle d_{\boldsymbol{\pi}'}, r_{\boldsymbol{\pi}'}\rangle| \tag{25}$$

$$\leq \frac{1}{1-\gamma}(|\langle d_{\boldsymbol{\pi}}, r_{\boldsymbol{\pi}}\rangle - \langle d_{\boldsymbol{\pi}}, r_{\boldsymbol{\pi}'}\rangle| + |\langle d_{\boldsymbol{\pi}}, r_{\boldsymbol{\pi}'}\rangle - \langle d_{\boldsymbol{\pi}'}, r_{\boldsymbol{\pi}'}\rangle|) \tag{26}$$

$$\leq \frac{1}{1-\gamma}(||d_{\boldsymbol{\pi}}||_1||r_{\boldsymbol{\pi}} - r_{\boldsymbol{\pi}'}||_\infty + ||d_{\boldsymbol{\pi}} - d_{\boldsymbol{\pi}'}||_1||r_{\boldsymbol{\pi}'}||_\infty) \tag{27}$$

$$\leq \frac{1}{1-\gamma}(2\epsilon + \frac{2\epsilon\gamma}{1-\gamma}) \tag{28}$$

$$\leq \frac{2\epsilon}{(1-\gamma)^2} \tag{29}$$

# D Experimental Details

## D.1 Introduction to the Environment

Kuhn Poker is popular research game, which is extensive-form and zero-sum with discrete observation and action space. There exists efficient min oracle with game tree search. For RoboSumo competition, both agents are multi-leg robots and observe the position, velocity and contact forces of joints in their body, and the position of their opponent's joints, which is much more challenging due to the high dimensional observation and action space.

## D.2 Feature Importantance in Robosumo Spider vs Spider

We show the important features in the Robosumo Spider vs Spider environment in Table 1.

| Original | Local | Global |
|----------|-------|--------|
| 38 | 32 | 31 |
| 32 | 43 | 39 |
| 33 | 38 | 29 |
| 43 | 37 | 43 |
| 41 | 31 | 41 |
| 29 | 35 | 34 |
| 35 | 42 | 42 |
| 39 | 41 | 35 |
| 42 | 40 | 37 |
| 26 | 28 | 44 |

Table 1: Feature Importance (Subset) in Robosumo Spider vs Spider. "Original": Index of the most important features of the victim's observation while playing against the attacker trained via the original self-play model. "Local": Index of the most important features of the victim's observation while playing against the attacker trained via our policy-coupled local attack method. "Global": Index of the most important features of the victim's observation while playing against the attacker trained via the global attack method as done in [6]

.

