# OpenReview forum: "Controllable Attack and Improved Adversarial Training in Multi-Agent Reinforcement Learning"
_NeurIPS.cc/2022/Workshop/TSRML — TSRML2022_

### Official Review · Reviewer_27hr · 2022-10-14
**Well-written and interesting**

**Overall Rating:** 7

**Summary:**

This paper studies the robustness of multiagent reinforcement learning in terms of adversarial agent. To ensure that the adversarial agent doesn't perform unstealthy and abnormal behaviors, the authors propose to constrain the state visitation distribution of the adversary, which is essentially restricting the policy distribution variation. The proposed timescale separation technique also improves the training convergence (which is essentially using different learning rate for the victim and the attacker).

**Strengths:**

Overall the paper is well-written and easy to follow. The proposed constraint to make the attack more stealthy is interesting. The theoretical analysis is sound, except one plausibly missing term as I point in the weakness section. The proposed timescale separation technique also seems to be effective in improving training convergence.

**Weaknesses:**

One thing that should be improved is the over claims or imprecise arguments. I listed a few as follows:

> "To provide provable defense" in the abstract.

I don't think the proposed adversarial training method can achieve provable defense against adversarial agents. Similarly, I don't think Corollary 1 (Robust Certificate) is a certification of attacks, because the provided bound is actually pretty loose ($\frac{1}{(1-\gamma)^2}$ is a very large number). By the way this corollary 1 also seems to be wrong by missing a maximum reward term $R_{max}$ (or the authors should explicitly assume that the reward range is from 0 to 1). Maybe the authors can check some references about certifiable robustness [1, 2] and then get better understanding of these terms. Corollary 1 is more like a worst-case bound in my opinion [3, 4].

> Line 41-43 as well as the abstract: "However, perturbing states or observations of an agent might not always be feasible/practical in real-life scenarios such as self-driving cars and robotic manipulation, where manipulating the observations may require attacking the communication systems which is hard to achieve".

I think the claim is imprecise because the perturbation of the observations is practical in the real-world because of the sensing noise and perception inaccuracies etc, as suggested by previous work [5, 6]. Particularly [6] has a discussion about this setting and some arguments in that paper might be useful to the authors. I would recommend rephrasing this sentence like "though the robustness against observational perturbation is important, the threat from adversarial agent also commonly exists in the real-world".
In addition, I believe that to some extent, we could also view the surrounding agent's adversarial behavior as the perturbation of the victim's observation space? Because from the perspective of the victim, the adversary's behavior is reflected by the changes of the observation.


Some other comments that might help to further improve this paper:

1.  I think the trick of using different learning rate is similar to the one in training GANs, which could be discussed or mentioned in the paper.

2. I suggest using consistent terms for the adversary and the victim, because sometimes they are called agent and sometimes they are called player, which is a bit confusing.

[1] Cohen, Jeremy, Elan Rosenfeld, and Zico Kolter. "Certified adversarial robustness via randomized smoothing." International Conference on Machine Learning. PMLR, 2019.

[2] Wu, Fan, et al. "Crop: Certifying robust policies for reinforcement learning through functional smoothing." arXiv preprint arXiv:2106.09292 (2021).

[3] Liu, Zuxin, et al. "Constrained variational policy optimization for safe reinforcement learning." International Conference on Machine Learning. PMLR, 2022.

[4] Achiam, Joshua, et al. "Constrained policy optimization." International conference on machine learning. PMLR, 2017.

[5] Zhang, Huan, et al. "Robust deep reinforcement learning against adversarial perturbations on state observations." Advances in Neural Information Processing Systems 33 (2020): 21024-21037.

[6] Liu, Zuxin, et al. "On the Robustness of Safe Reinforcement Learning under Observational Perturbations." arXiv preprint arXiv:2205.14691 (2022).

**Overall Recommendation:**

Overall I think this is an interesting paper with good theoretical and empirical justification. Some comments for improving this paper are listed above.

**Review Confidence:**

4: The reviewer is confident but not absolutely certain that the evaluation is correct

---

### Official Review · Reviewer_jQ6b · 2022-10-18

**Overall Rating:** 7

**Summary:**

The authors propose an adversarial attack in the RL setting where the attacker budget is limited in terms of difference in state visitations, making the adversarial agent behave more similar to a benign agent to the human eye. They show that they can guarantee such behaviour by bounding the adversarial policy's total variation distance, which they can in turn enforce by reparameterizing their policiy as a mixture of an unconstrained adversarial policy and the original benign policy. Additionally, they propose a defence strategy for an unconstrained attack, which, for zero-sum games, approximately yields the NE while enabling efficient training via time-scale separation. They demonstrate both the improved efficiency of their attack and the effectiveness of their defence empirically.

**Strengths:**

* They consider the important setting of adversarial agents in RL which, to a human eye, behave similarly to benign agents.
* They propose a simple yet very effective approach on how to limit the attack budget in terms of state visitation difference.
* Empirically, their novel attack seems to be more efficient than unconstrained attacks, yielding much smaller changes in agent behaviour at the same effectiveness.
* Empirically, their defence seems to stabilise training and make it tractable via time-scale separation.

**Weaknesses:**

* Effectiveness of the proposed attack is unclear and not discussed beyond Corallary 1. E.g. how closely do we match this bound?
* A more quantitatively rigorous analysis  (e.g. KL divergence) of empirical results on efficiency of the proposed attack would be helpful. Further how was the winning rate of the unconstrained attack fixed/reduced?
* There is a very large number of typos (see minor comments) that should be fixed for a potential camera ready version.

### Minor Comments
* Typos "an" vs "a", (e.g. line 7), line 85 \mu => \nu, Line 163 **A**lgorithm 1
* \citet vs \citep (e.g. line 38)
* Epsilon is used as attacker budget in terms of total variation distance and state visitations (l1-norm), making later references ambiguous.
* Notation of $\pi_\nu(\theta)$ not ideal. Perhaps $\pi_\nu^*$  would be a good choice.
* Basic, unconstrained attack in L149 should be $\epsilon=1.0$
* Plots on page 5 and 6 make paper very slow to load and are readable only under extreme zoom. Further, visualising a rolling mean and variance in Figure 3 would make the plot much more readable.

**Overall Recommendation:**

The authors propose a novel (to the best of my knowledge) attack class for adversarial agents in the RL setting, an efficient approach to training a corresponding malicious agent, and a defence strategy against an unconstrained agent. While the presentation of the work could (and should in my opinion) be improved (more analysis, fix typos, better plots, add conclusion/discussion), I believe the technical insights to be interesting enough to outweigh these flaws. Nevertheless, I highly recommend the authors to improve the presentation for the next version of the paper.

**Review Confidence:**

3: The reviewer is fairly confident that the evaluation is correct

---

### Official Review · Reviewer_Ci6F · 2022-10-21
**Interesting paper that constructs a bounded attack in multi-agent RL setting and improves adversarial training. Experimental section slightly unclear**

**Overall Rating:** 8

**Summary:**

This paper constructs a bounded attack in multi-agent reinforcement learning that lead to imperceptible differences in the state visitation distribution of agents and thereby such attacks could be undetectable. The authors also propose an adversarial training scheme to make the victim agents robust to attacks via minimizing their exploitability using a timescale separation algorithm that leads to guaranteed convergence. The ideas are verified on two zero-sum games via empirical results.

**Strengths:**

- The paper makes a valid contribution by pointing out that constrained attacks that bound the policy difference between agents in the sense of total variation lead to imperceptible attacks where the state distribution shift is not prominent when compared with unconstrained attacks by an adversarial agent which could then be easily detected.

- The paper also proposes a two timescale based adversarial training with the goal of minimizing the exploitability of agents and show that it leads to better convergence and closest performance (numerically) with the min-oracle approach.

**Weaknesses:**

- Are the rewards bounded? In (27), is it assumed that $||r_{\pi’}||_{\infty} = 1$? Otherwise the bound in Corollary 1 is not clear
- Is it possible to interpret the bound further in Theorem 1? $\epsilon$ could be a very large value or the discount factor could be very close to 1 in which case the bound becomes trivial.
- Was only 1 instantiation/realization of the policy picked in the numerical result for Fig.1 with same win rate for constrained and unconstrained attacks? -What happens on average?
- Where is proposition 3? It is mentioned on Line 141 on Page 4.
- There is a technical problem with the paper PDF. Firstly, the size is ~35 MB and it always crashes no matter the environment it is opened in. This seems to be due to the graphics.
- The figures contain a lot of information but the font on them is simply unreadable.
- In Fig. 2, it is difficult to verify the author’s statement that 2-timescale separation with largest ratio comes closest to the min oracle. Also, please label the sub-figures-does each plot correspond to a single seed?


**Overall Recommendation:**

Well-written paper with important contributions. Experimental section could be vastly improved.

**Review Confidence:**

3: The reviewer is fairly confident that the evaluation is correct

---

### Decision · Program_Chairs · 2022-10-23

**Decision:**

Accept

**Comment:**

Following the unanimous recommendations from reviewers, the submission is accepted.